# Learning object representations through amortized inference over probabilistic programs

**Francisco Silva**  *francisco.c.silva@inesctec.pt*
*INESC TEC - Institute for Systems and Computer Engineering, Technology and Science*
*Faculty of Sciences, University of Porto*

**Hélder P. Oliveira**  *helder.f.oliveira@inesctec.pt*
*INESC TEC - Institute for Systems and Computer Engineering, Technology and Science*
*Faculty of Sciences, University of Porto*

**Tania Pereira**  *tania.pereira@inesctec.pt*
*INESC TEC - Institute for Systems and Computer Engineering, Technology and Science*
*Faculty of Engineering, University of Porto*

**Reviewed on OpenReview:** *https://openreview.net/forum?id=nUFSrlJaUr*

## Abstract

The recent developments of modern probabilistic programming languages have enabled the combination of pattern recognition engines implemented by neural networks to guide inference over explanatory factors written as symbols in probabilistic programs. We argue that learning to invert fixed generative programs, instead of learned ones, places stronger restrictions on the representations learned by feature extraction networks, which reduces the space of latent hypotheses and enhances training efficiency. To empirically demonstrate this, we investigate a neurosymbolic object-centric representation learning approach that combines a slot-based neural module optimized via inference compilation to invert a prior generative program of scene generation. By amortizing the search over posterior hypotheses, we demonstrate that approximate inference using data-driven sequential Monte Carlo methods achieves competitive results when compared to state-of-the-art fully neural baselines while requiring several times fewer training steps.

## 1 Introduction

Neurosymbolic models have recently re-emerged from the need to extract and incorporate *a priori* symbolic information while processing high-dimensional and unstructured images, bridging pixel-level transformations modeled by neural networks and symbols easily interpretable to humans (Garcez & Lamb, 2023; Chaudhuri et al., 2021). This goes in line with the idea that incorporating symbolic knowledge, along with the information they carry by grounding them in world entities, should make the generalization power of machine learning systems — in particular neural networks — closer to what humans achieve (Lake et al., 2015; 2017; Goyal & Bengio, 2022; Garcez & Lamb, 2023). Even though recent developments have been made (Mao et al., 2019; Feinman & Lake, 2021; Liang et al., 2022), the task of learning object representations is still mostly tackled with no significant high-level symbolic constraints on pattern recognition systems, which makes poor generalization expected (Goyal & Bengio, 2022; Lake et al., 2017). Moreover, the natural existence of uncertain choices is not addressed in those systems (Pearl, 1988).

The capacity to individuate and track objects over time is a fundamental human ability that is demonstrated since early infancy, following intuitive spatiotemporal constraints and progressively learning what could be expected from their physical interactions (Spelke, 1990; Carey, 2009; Spelke & Kinzler, 2007; Baillargeon

et al., 1985; Lin et al., 2022; Ullman et al., 2017). Such *core knowledge* could be represented as rich generative models that, innately given or not, humans use to not only make predictions about possible future states but also to efficiently infer explanations for what is being observed (Lake et al., 2017; Ullman et al., 2017; 2018; Ullman & Tenenbaum, 2020). Within the view of *vision as inverse graphics*, inferring explanatory information from an observed image is seen as inverting the generative model that produced such an image (Stuhlmüller et al., 2013; Horn, 1977). We then propose a model for inference compilation through slot-attention (ICSA), a neurosymbolic system that approximates symbolic object properties by fast amortized inference in a probabilistic program of scene generation. Unlike previous works on this line (Kulkarni et al., 2015; Jampani et al., 2015), we combine stronger inductive biases on the way features get extracted from the observed images to improve posterior distribution proposals provided by neural networks. In particular, we adapted the SA-MESH algorithm (Zhang et al., 2023) as the neural encoder responsible for extracting object-centric features that are further used for sequential inference of generative variables (Le et al., 2017; Paige & Wood, 2016). Moreover, we do not rely on pre-trained architectures for object detection, as our main objective consists of implementing a system that learns object-centric representations (akin to *object files* (Kahneman et al., 1992; Green & Quilty-Dunn, 2021; Stavans et al., 2019)) using symbolic *a priori* constraints on what kind of features should be captured from pixel-level patterns. We evaluate our model in the main downstream task of set prediction against fully neural baselines. The ICSA generative model and inference networks were implemented using the Pyro probabilistic programming language (Bingham et al., 2019).

**Contributions**   The main contributions of our work are as follows: (1) the application of a data-driven posterior proposal module with strong object-centric inductive biases for more interpretable representations learning; (2) measuring the advantageous of having richer prior knowledge - in the form of a fixed generative model - for learning object-centric representations in terms of data efficiency, while accounting for posterior uncertainty and (3) an experimental setup that evaluates the behavior of object-centric models in different but plausible out-of-distribution inference regimes.

## 2   Related work

### 2.1   Probabilistic programming for computer vision

Probabilistic programs specify *a priori* knowledge about explanatory variables of a generative model, along with the dependencies between them using the `sample` primitive command (van de Meent et al., 2018; Gordon et al., 2014). Probabilistic programming languages (PPL) also implement inference engines to infer the probabilities of latent variables given the observed data, using a variety of algorithms such as methods within the family of Markov chain Monte Carlo (MCMC), stochastic variational inference (SVI), and amortized inference. Universal PPL, which allow the construction of stochastic and unbounded generative programs (Goodman & Stuhlmüller, 2014), are also capable of incorporating neural networks-based operations to increase inference speed (Bingham et al., 2019; Lavin et al., 2021), which makes their application in computer vision tasks more accessible. For instance, learning the parameters of an interpretable probabilistic program of image generation reduces to learning the bottom-up transformation from pixel data to each random variable intended to be inferred from the scene. Several works have already followed this line, motivated by the idea of making MCMC proposal steps based on data-driven functions (Kulkarni et al., 2015; Jampani et al., 2015; Mansinghka et al., 2013; Ullman et al., 2018; Yuille & Kersten, 2006; Wu et al., 2015). These ideas are also convergent with some research lines from neurosymbolic modeling, by exploring the benefits of placing stronger top-down influences (with interpretable symbolic programs) over the kinds of features that can be extracted from images with neural networks (Garcez & Lamb, 2023; Chaudhuri et al., 2021; Goyal & Bengio, 2022).

### 2.2   Inference Compilation

Inference compilation (IC) consists of training specialized neural networks to propose distribution parameters for an arbitrary generative probabilistic program (Paige & Wood, 2016; Le et al., 2017; Harvey et al., 2019; Baydin et al., 2019), combining the advantages of sequential inference methods (Doucet et al., 2001b;

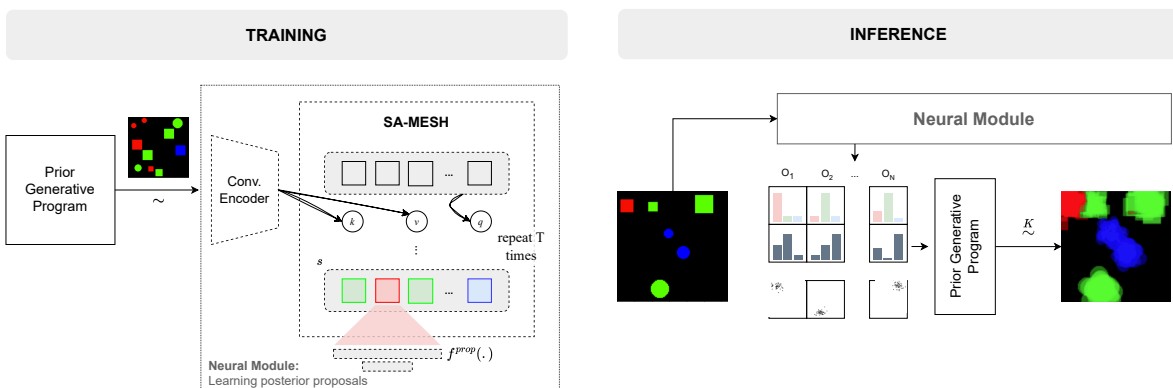

Figure 1: An overview scheme of ICSA, starting with a prior generative model implemented as a probabilistic program, which samples object latent variables from broad uniform prior distributions and generates batches of training scenes. A neural module consisting of a convolutional encoder and the SA-MESH attention algorithm is responsible to learn object-centric representations in the format of a set of slots $\mathbf{s}$, which are further used by specialized networks $f^{\mathrm{prop}}(.)$ to output the parameters of the posterior distributions of each latent variable. At inference time, a test scene is observed and $K$ samples are drawn from the proposed posterior distributions, being then evaluated using importance sampling (ICSA-IS), or an object-wise SMC method referenced as score-resample (ICSA-SR), which scores and resamples objects iteratively until the whole scene gets explained.

2009) and amortized data-driven inference (Gershman & Goodman, 2014; Jampani et al., 2015). Before observing any target dataset where inference is intended to be performed, IC works by learning distribution proposals for a prior generative model. After training, the inference system observes a set of unseen data and reuses the learned proposals to approximate the true posterior in a sequential importance sampling strategy. Even though the IC algorithm has already been used to solve vision tasks (e.g. captcha solving (Le et al., 2017)), a strong application has been focused on bridging probabilistic programming and large-scale scientific simulators, allowing a significant improvement in inference efficiency (Baydin et al., 2019; Munk et al., 2019; Lavin et al., 2021; Munk et al., 2022).

## 2.3 Object-centric representations

The task of learning object representations has a long history in computer vision. Designing systems capable of individuation objects in a scene is a challenge transversal to several downstream applications (e.g. object categorization or pose estimation). However, learning object representations is frequently addressed without any explicit purpose (e.g. (Locatello et al., 2020; Feinman & Lake, 2021; Eslami et al., 2016; Kosiorek et al., 2019; Seitzer et al., 2022; Anciukevicius et al., 2022)), hence no supervision is provided, hoping to learn general representations. Recently, the slot attention (SA) (Locatello et al., 2020) method was introduced and has served as the foundation for several other approaches for learning object representations (Sajjadi et al., 2022; Elsayed et al., 2022; Kipf et al., 2021; Wu et al., 2022; Seitzer et al., 2022; Singh et al., 2022; Zimmermann et al., 2023; Biza et al., 2023; Webb et al., 2023; Mansouri et al., 2023; Zhang et al., 2023; Brady et al., 2024; 2023; Wiedemer et al., 2023; Kori et al., 2024). SA conceptually emulates a system of *object files* (Kahneman et al., 1992; Carey & Xu, 2001; Xu et al., 1999), where separated representations (slots) bind to specific regions of the input image via attention mechanisms, aggregating information about them. This large body of work has followed the idea that inductive biases injected as architectural constraints should restrict the vast combinatorial possibilities that a scene's latent representation might navigate through (Bengio et al., 2013; Goyal & Bengio, 2022), opening doors for these structured representations to be used in different downstream tasks.

# 3 Methods

Our proposed system follows the ideas from the *analysis-by-synthesis* paradigm and the *Helmholtz machine* (Hinton et al., 1995; Dayan et al., 1995), with posterior hypotheses being computed by fast bottom-up neural modules, and weighted by comparing the corresponding hypothetical scene against some observation. ICSA combines a probabilistic program that samples a set of properties associated with an unknown number of objects to generate a scene, with an inference module that learns object-centric representations and sequentially proposes parameters for the posterior distribution of each latent variable encountered in the program's execution trace. We further describe in detail each module, starting with the generative program (Sec. 3.1) and then the main components of the implemented inference strategy based on the IC algorithm (Paige & Wood, 2016) (Sec. 3.2).

## 3.1 Generative model

We formalize our generative model as a latent variable model of ground-truth latent variables $\mathbf{z} \sim p(\mathbf{z})$, which generates image samples $\mathbf{x}$ in the form of object scenes. The generative model samples object attributes from prior distributions over pre-specified and fixed libraries of possible values. Observed images are rendered through a non-differentiable generator $g : \mathcal{Z} \to \mathcal{X}$, which maps a set of latent variables $\mathbf{z}$, with training support $\mathcal{Z}$, to an image $\mathbf{x}$, i.e. $g(\mathbf{z}) = \mathbf{x}$. Since there are no causal dependencies between latent variables, $p(\mathbf{z})$ factorizes in $\prod_i p(z_i)$.

The number of objects present at each scene is determined by a Bernoulli mask array with success probability $p_M$. In our experiments, we set $p_M$ at 0.5 and generate scenes with up to 10 objects. We assume uniform priors over all discrete and continuous variables, no occlusions, and objects always fully appear within the observation canvas. For instance, shape is distributed between the categories *ball* and *square*; size between *small*, *medium* and *large*, with associated diameters set at 10, 15 and 20px, respectively; and color between *red*, *green* and *blue*.

For simplicity, $g$ was implemented using the Python Pillow imaging library, generating a batch of images at each execution trace during the training phase.

## 3.2 Training

The implemented inference network consists of a slot-based feature extractor, adapted from (Zhang et al., 2023), optimized by an IC algorithm that sequentially learns data-driven parameter proposals for the posterior distribution of latent variables. We further detail these main components over the next subsections.

### 3.2.1 Learning object-centric representations

We adopted the SA-MESH architecture proposed in (Zhang et al., 2023), which connects the cross-attention mechanism implemented in previous SA works (Locatello et al., 2020; Kipf et al., 2021; Biza et al., 2023; Wang et al., 2023) with optimal transport, by minimizing the entropy of the attention mask that results from applying the Sinkhorn algorithm (Cuturi, 2013; Sinkhorn & Knopp, 1967) over the computed distance matrix. The core advantage over the standard SA approach is that, by making the slots $\to$ inputs mapping *exclusively* multiset-equivariant instead of only set-equivariant (as in SA), the attention maps become sparser and slot representations are less prone to collapse when objects are identical. We provide more details on this in App. A.1.

### 3.2.2 Fixing symmetries at inference time

Since slots are randomly initialized and proposals for posterior distributions of latent variables are assumed to be permuted by any arbitrary object-wise order during training, there are no guarantees that, at inference time, the same order will be followed for every inference particle (since each one represents an independent run of the inference network). More concretely, fixing no structure to the slot $\leftrightarrow$ object binding would result in an incoherent evaluation of the hypotheses raised for a certain individual object. Following a similar method proposed by Le et al. (2017), we find a permutation $\pi^k : \{1, ..., N\} \to \{1, ..., N\}$, associated with

the $k$-th proposed traced, such that Equation 1 is guaranteed, being $r$ the inferred Euclidean distance of each object to the origin of the canvas. Note that, since each trace orders objects arbitrarily, $\pi^i$ and $\pi^j$ are independent permutations $\forall i, j \in [K]$, $i \neq j$.

$$r_{\pi^k(1)} \leq r_{\pi^k(2)} \leq \cdots \leq r_{\pi^k(N)} \tag{1}$$

This way, proposals for individual objects can be weighted independently and sequentially given the sampled choices made by each trace, up to ambiguities raised by this ordering choice, which will be later discussed.

### 3.2.3 Objective function

As shown in Fig. 1, specialized neural predictors $f^{\text{prop}}(.)$ are used to estimate the parameters of posterior distributions regarding each latent variable, taking the set of slots $\mathbf{s}$ as input. For instance, $f^{\text{prop}}_{\text{shape}}(\mathbf{s})$ parameterizes the posterior probabilities for each object's shape.

In IC, the optimization direction is guided by a Kullback-Leibler (KL) divergence, which measures the "distance" between the approximated and the real posterior distributions. In contrast with the KL term traditionally implemented in variational autoencoders (VAE) (Kingma & Welling, 2013) and similar models, the arguments of this KL divergence are flipped. The loss function used is expressed in Equation 2. Note that the expectation is computed with samples drawn from the generative model $p(\mathbf{x}, \mathbf{z})$, and not using the approximated distribution $q(\mathbf{z}|\mathbf{x}; \phi)$ for latent variables sampling (as in VAE). This training mode is similar to the "sleep" phase of wake-sleep algorithm (Dayan et al., 1995), thus training the neural artifact only requires a fully specified generative model to sample from (see in Equation 2), and no offline training dataset (Le et al., 2017). By inverting the arguments, minimizing the inclusive KL divergence encourages the approximate posterior to distribute its density to cover the entire support of the true posterior, which brings benefits when the variational distribution tries to approximate complex and multimodal true posteriors (for more in-depth information about this behavior and the loss formulation, please refer to (van de Meent et al., 2018)).

$$\begin{aligned} \mathcal{L}(\phi) &= \mathbb{E}_{p(\mathbf{x})}[D_{\text{KL}}\left(p(\mathbf{z}|\mathbf{x})||q(\mathbf{z}|\mathbf{x}; \phi)\right)] \\ &= \mathbb{E}_{p(\mathbf{x}, \mathbf{z})}[-\log q(\mathbf{z}|\mathbf{x}; \phi)] + \text{const} \end{aligned} \tag{2}$$

### 3.3 Inference through Sequential Importance Sampling

We employed a sequential Monte Carlo (SMC) (Chopin et al., 2020; Doucet et al., 2001a; Le et al., 2017; van de Meent et al., 2018) method to approximate posterior inference over the specified generative program. After aligning proposal traces, data-driven posterior hypotheses $[\hat{z}^k_{1:N}]^K_{k=1}$ are proposed for the set of $K$ particles (i.e. each of the $K$ samples drawn from the proposed posterior), which are evaluated and resampled in an object-wise sequential manner (Gothoskar et al., 2023). More concretely, at the first inference step, only the first object — i.e. its associated latent variables — is sampled, scored and resampled. Following this, more objects are iteratively added until the observed scene is fully explained. For each proposed object $o_j$ such that $j \in \{1, 2, \ldots, N\}$, the importance weight for the $k$-th particle, $w^k_{o_1:o_j}$, is computed as following:

$$w^k_{o_1:o_j} = \frac{p(\hat{z}^k_{o_1:o_j}, \hat{\mathbf{x}})}{q(\hat{z}^k_{o_1:o_j}|\mathbf{x})} , \tag{3}$$

where $\hat{z}_{o_1:o_j}$ comprises the sampled values for the latent variables associated with the set of objects $\{o_1, \ldots, o_{j-1}, o_j\}$, being $\mathbf{x}$ and $\hat{\mathbf{x}}$ the observed and generated scenes, respectively.

As shown in Equation 3, the likelihood of the resultant image in the $k$-th proposed trace, when proposing latent variables for object $o_j$, is evaluated against the full observed image $\mathbf{x}$. This means that, instead of only scoring how well the generated object explains the observed one at the same location, we also evaluate the impact of accepting the proposed object while aiming to explain the whole scene. Image likelihoods are computed using a continuous observation model modeled by a Gaussian distribution with a fixed noise

parameter $\sigma_L$, $\mathcal{N}(\mathbf{x}; \hat{x}_{o_1:o_j}^k, \sigma_L)$. Here, $\hat{x}_{o_1:o_j}^k$ represents the scene that is produced when rendering objects 1 to $j$ according to the proposed attributes inferred in particle $k$. We further discuss the impact of this inference strategy (Sec. 6).

## 4 Experiments

We first validate our approach within the generative environment specified by our model (Sec. 3.1) under the set prediction task (Sec. 4.1) and directly compare its performance and data-efficiency against SA-MESH (Zhang et al., 2023). Then, we further evaluate the application of our model in a scaled data complexity environment, using the CLEVR dataset (Johnson et al., 2017) [1]. For the first case, we generated a test dataset of 500 scenes uniformly distributed over the number of sampled objects so that we could measure possible failing modes that become more evident when the number of objects to be predicted grows. The main assessment point is to effectively compare the performance of data-driven inference proposals evaluated through Monte Carlo methods (either with an object-wise sequential procedure or by weighting and resampling full posterior proposals) against posterior predictions learned in a discriminative way by fully neural models.

### 4.1 Set prediction

The set prediction task involves taking the set representations (slots) and computing the target attributes for each object, allowing any arbitrary order among them. The main goal is then to find a unique match between the predicted and the ground-truth sets of object attributes, validating the learned slot representations.

**Baseline**  We compared ICSA performance against SA-MESH (Zhang et al., 2023), since we adopted its slot-based image encoding approach in our inference procedure. Hence, a direct comparison in terms of performance over data efficiency can be made. To train SA-MESH in the same data domain, we employed our prior generative program (Sec.3.1) to generate an offline dataset with 30K images.

**Results**  Following previous works (Locatello et al., 2020; Zhang et al., 2019; 2023), we computed the average precision (AP) values over different distance thresholds, using the same hold-out test set, for an increasing number of training steps to assess training efficiency. Fig. 2b shows AP values computed over all scenes with $N \in \{1, \ldots, 10\}$ objects for SA-MESH and ICSA. We split the ICSA results in two different inference procedures, where ICSA-SR denotes the score-resample procedure and ICSA-IS refers to the application of the traditional importance sampling algorithm. These different inference procedures within ICSA allow to investigate the trade-off between having a more structured method that enhances interpretability but raises some ambiguities, especially with larger $N$ values (ICSA-SR), and a faster but less interpretable method that represents posterior hypotheses as full program traces (ICSA-IS).

The AP curves over increasing training requirements (Fig. 2b) show that ICSA models achieve at least the same level of prediction performance as SA-MESH, with $\approx 20$ times fewer training steps. Comparing both ICSA inference methods, approximating the posterior with a single-step proposal weighting (ICSA-IS) allowed to achieve higher AP values, even though not holding for $\sigma = 0.0625$. On the other hand, predictions made through ICSA-SR achieve a higher ratio of true positives for lower distance thresholds, which can be explained by the sequential characteristic of its inference process that enables a thorough evaluation of the object-wise proposals. However, pre-establishing an order for the set of objects can raise a failing mode where objects are not discovered due to ordering ambiguities (see Fig 13). Detailed comparative plots of AP values for each group of scenes can be found in App. A.2.

Fig. 2a shows examples of the sequential process that occurs during inference with the score-resample algorithm: for each object, the best proposal is resampled given how well it contributes to explaining the initial observation. This is why object overlaps tend to be avoided. Even though overall AP values obtained in ICSA-IS are higher than by going through the sequential score-resample algorithm, the final predictions may suffer from objects overlap (see Fig. 2c), which is something that the ICSA generative model does not

---

[1]The implemented code and test samples are available at `https://github.com/franciscocms/ic-slotatt`.

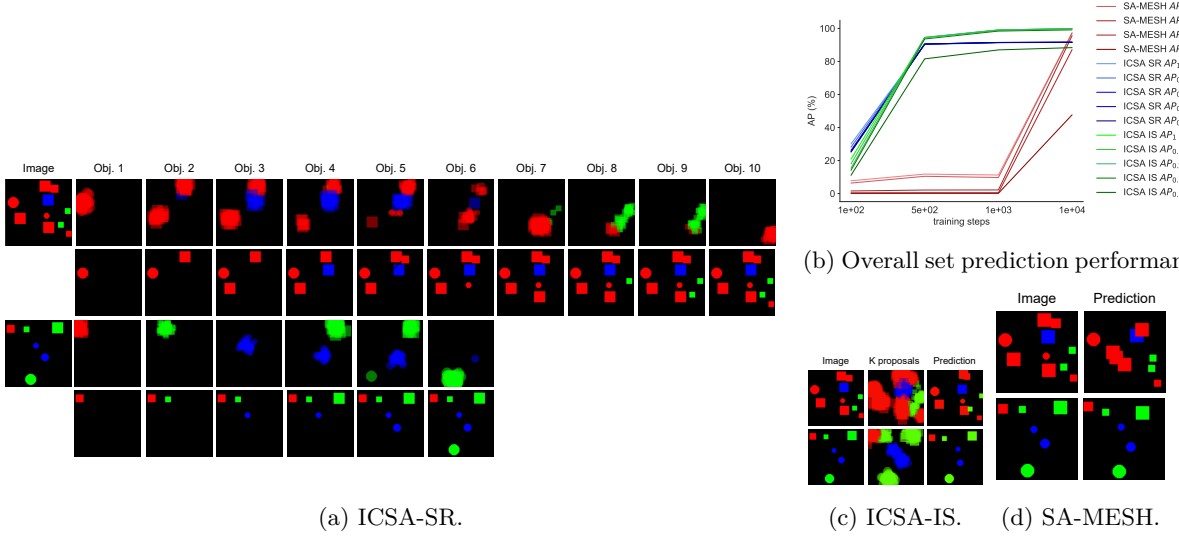

(a) ICSA-SR.

(b) Overall set prediction performance.

(c) ICSA-IS.     (d) SA-MESH.

Figure 2: **(a)** Sequential inference with ICSA-SR, where at each step, proposals for an additional object (top row) are weighted and resampled (bottom row). **(b)** Comparative results for set prediction performance expressed in AP (%), for different training requirements, over 5 random seeds, averaged over all scenes with $N \in \{1, ..., 10\}$ objects. Red lines correspond to inferences made with SA-MESH, while green and blue lines correspond to the ICSA with importance sampling and score-resample inference procedures, respectively. Darker lines are associated with lower distance thresholds $\sigma$. **(c)** Inference with IS by evaluating whole-scene proposals. **(d)** Predictions obtained by running the SA-MESH model and rendering the set of inferred object properties. For all inference alternatives, two random test examples (with 10 and 6 objects, respectively) are shown to illustrate overall results. The model parameters used to compute these examples were loaded after 10K training steps for a fair comparison between SA-MESH and ICSA inference.

generate *a priori*. Object properties inferred by SA-MESH also suffer from the same issue, which becomes more evident when the number of objects present in the scene grows (see Fig. 2d and Fig. 12c). Finally, at an image level, SA-MESH results tend to look worse than the ones obtained with ICSA inference models due to (1) all models were trained for set prediction tasks and not for object discovery (i.e., with an objective function that does not operate on pixel-level errors) and (2) SA-MESH does not account for uncertainty in its predictions. This is quantitatively shown in the performance decrease for lower $\sigma$ values (see Fig. 2b). We provide additional inference details and ablation studies in App. A.2.

## 4.2 Model misspecification scenarios

We specified our generative program with broad and uniform priors, being a reasonable strategy to avoid not covering possible posterior regions (Le et al., 2017; Paige & Wood, 2016; Gothoskar et al., 2021). However, we also investigated the behavior of our inference methods under some out-of-distribution (OOD) situations, aiming to explore what could be expected from a misspecified model that tries to infer object properties in data that it cannot generate *a priori*. Here, we focused on OOD scenes that contain objects with properties never seen in training, either by adding a new shape to the test library, a new color, or both at the same time. As shown in Fig. 3a, the test sample in the first row contains green triangles, while the second image shows two objects with a new randomly sampled RGB code — for instance, $(150, 56, 195)$. For both examples in this figure, we sampled object positions and these remained fixed.

We also explored more challenging OOD samples, generated by sampling object properties over distributions broader than the ones covered by our prior generative program, and composing them to generate objects with unseen shapes and colors at the same time (see Fig. 3b). More concretely, we added different geometries to the shape library (ellipses with varied levels of eccentricity, non-squared rectangles, and triangles), and all colors were sampled from uniform distributions over the full range of RGB codes. To facilitate the

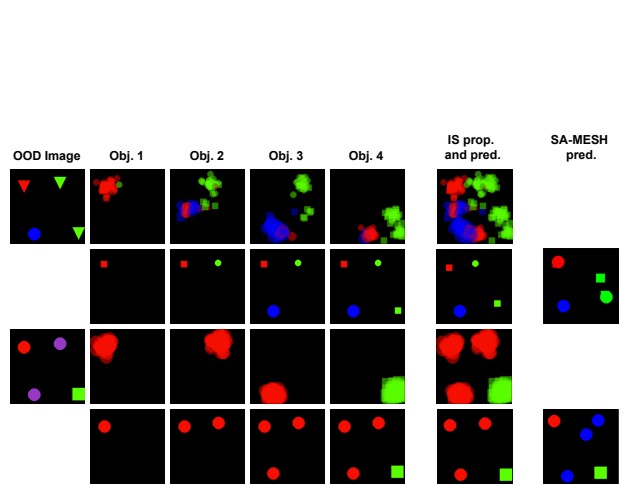

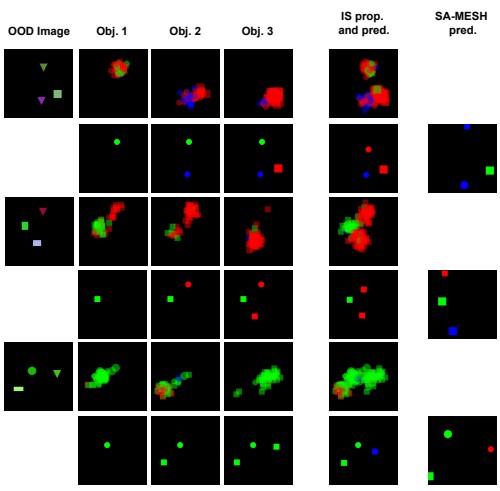

(a) Color and shape OOD scenarios generated separately.

(b) Color and shape OOD scenarios generated at the same time.

Figure 3: **(a)** Out-of-distribution samples to represent misspecification behavior at different situations (shape in the first OOD image, and color in the second). Property inference and generated samples are shown for ICSA-SR (left), ICSA-IS (middle) and SA-MESH (right). **(b)** More complex out-of-distribution samples and inference results obtained with ICSA-SR (left), ICSA-IS (middle) and SA-MESH (right) inference procedures. The traces overlay obtained for each object in ICSA-SR are shown in the same row of the observed sample, while the corresponding final sampled trace is shown in the row below.

interpretation of these results, we chose to maintain the number of objects on these scenes relatively low, and positions were sampled so that ordering ambiguities could not be an issue *a priori*.

**Results** In the most simple cases of Fig. 3a, it seems like both ICSA methods hold results that are more coherent and close to the observed objects, even when not capable of generating them (note that the generative program remained the same for all approaches, hence OOD latents could not be inferred neither generated). We would like to highlight the fact that the number of objects that are predicted corresponds to the reality in ICSA posterior proposals, in contrast with SA-MESH. More, even though objects are not located in positions that could cause ordering ambiguities (which is seen in the traces of the second OOD image), uncertainty is easily observed in the shape misspecification example where slots showed difficulties binding to objects exclusively. This was not the case for the color misspecification scene, which might indicate that pixel-level properties are exploited at different extensions when the attention mechanisms of the neural module are optimized. In both examples, the SA-MESH baseline detected additional objects in the scene, and uncertainty is not as easily observed and qualified as with ICSA probabilistic inference. In the cases of Fig. 3b, conclusions are less trivial to be drawn. However, the predictions made by ICSA (mainly with SR inference) seem to be more consistent when multiple OOD scenarios occur in the same observation, making more coherent shape inferences for objects with similar shapes even if these don't belong to its prior library. In SA-MESH, inferred properties are less stable and some inconsistencies seem to occur frequently (e.g. predicting a different color for objects that are presented with very similar colors, as seen in the middle and bottom examples).

### 4.3 Scaling data complexity

We also tested ICSA in a data environment with increased scene complexity: the CLEVR dataset (Johnson et al., 2017), a synthetic dataset of 3D scenes of up to 10 objects, varying in size, shape, color and material. Considering the different group of generative variables required to explain CLEVR scenes, we modified our prior generative program to include them. To speed up inference times, only the ICSA-IS inference strategy

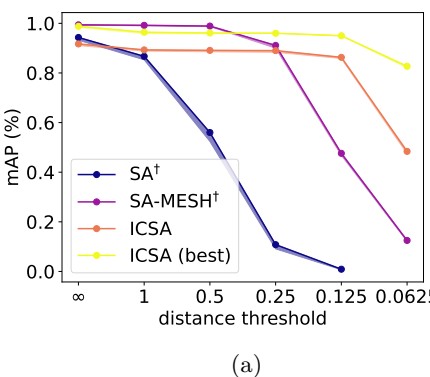 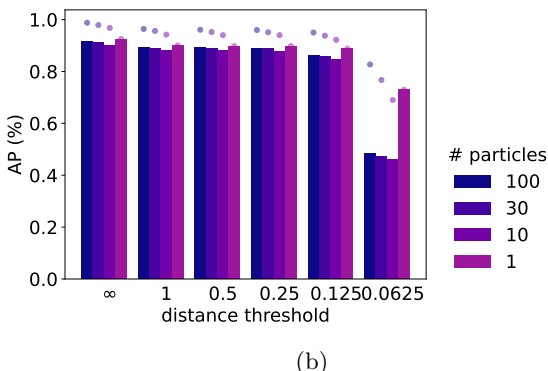

(a)  (b)

Figure 4: **(a)** Set prediction results measured as mAP in % (values are presented as mean with shaded areas representing associated standard deviations) for different distance thresholds $\sigma$. Results obtained with SA and SA-MESH (with $^\dagger$) were copied from the respective papers. **(b)** Particle ablation results of ICSA, considering $K \in \{100, 30, 10, 1\}$, with best-particle values represented as dots on top of each bar.

is used to compute weights for posterior proposals and further mAP values. This makes the computational cost much lower since it requires only the generation of $K$ predicted hypotheses for each test scene, instead of $K \times N$, being $N$ the number of predicted objects.

We provide these results mainly for a matter of empirical practicality test, demonstrating that ICSA scales for more complex data environments, assuming the generative model is known *a priori*. Given our focus on training efficiency, we constrained training duration to the same 10K steps as in the previous experiments, which we believe already allows us to draw some conclusions compared to neural baselines.

**Results**  We show the achieved mAP values for ICSA and baseline models in the CLEVR validation dataset in Fig. 4a. As mentioned above, the main purpose of this scaling experiment is to demonstrate that our method is capable of learning to invert more complex generative models, assuming knowledge about them. However, looking at these quantitative metrics, we must also address the main explanations for why ICSA is not as accurate as SA-MESH, at least regarding higher distance thresholds. First, to speed up ICSA training, we chose to decrease some rendering parameters related to image quality, which makes training and validation images different in terms of general image quality in a way that may not be negligible. The second reason is related to the misalignment between pixel-level likelihood metrics (which affect particle weights for resampling) and latents prediction correctness, meaning that not always the trace that better predicts the set of object properties will be the resampled one — for instance, we illustrate such an example in Sec. 4.4. As defined in Sec. 3.3, scoring each particle involves evaluating the observed scene under a Gaussian distribution with an uncertainty parameter $\sigma_L$. For this uncertainty calibration, we used the effective sample size (ESS) metric to find $\sigma_L$ values such that the ratio $\frac{\text{ESS}}{K}$ would lie around 0.1, aiming to mitigate this misalignment issue by not having too many hypotheses equally possible of being resampled. We also plot the mAP values have the best particle — found by comparing mAP values *a posteriori* — been resampled for each validation scene (ICSA (best)), which clearly shows the effect of this misalignment. In fact, we see that if pixel-level likelihood scores were paired with property predictions, ICSA would deliver very competitive results. In Fig. 4b, we ablate the number of particles used for inference, showing that better results don't necessarily come from having more hypotheses to choose from. Again, we show that better hypotheses could have been sampled from the proposed posterior, but having a single particle seems to yield the best results because it avoids particle scoring issues. Finally, we measure training efficiency by comparing the number of batches used to run forward and backward for each approach, as well as their size. Considering that we were not able to find the number of training steps employed for the set prediction task with CLEVR in SA-MESH, we used the values reported by Locatello et al. (2020) in the original SA paper, which were 150K training steps with batches of 512 scenes — which is why we added SA results to Fig. 4a. In contrast, our experiments with CLEVR used 10K training steps, with a batch size of 64. This represents a proportion of $120\times$ less images seen during ICSA training.

### 4.4 Measuring uncertainty in ICSA

Considering the particle-based nature of inference procedures in ICSA, it is possible to observe and measure prediction uncertainty under challenging conditions, e.g. comparing occlusion and no-occlusion pairs of identical scenes, revealing how the prediction of different visual properties is affected in such situations. Fig. 5 illustrates two examples of an identical occlusion scenario, differing in the similarity of the colors of the occluder and the hidden object. In the first case (Fig. 5a), some uncertainty can be observed in all properties, but the fact that each object presents a unique color makes it easier to coarsely distinguish shape and materials (first two bar plots). Also, it is interesting to note that even though only a "small" object could be occluded in that situation, the correct size was only inferred with low uncertainty once the occlusion event ended. When objects present the same color (Fig. 5b), higher uncertainty can be observed (e.g. when inferring object materials), but these are also resolved once objects are no longer occluded. In the selected case, the shape of the occluded object remained uncertain, but here occlusion is no longer a plausible explanation for such behavior.

**Resampling mistakes** As mentioned before, a possible explanation for incorrect prediction lies on a misalignment between the likelihood weights computed for each particle and the probability distributions that actually compose the inferred posterior. For instance, in the shape uncertainty case that is shown in Fig. 5b, the likelihood weights of all inference particles are distributed as shown in Fig. 5c, aggregated by the sampled shape class. It is clear that, in some cases, these likelihood weights are not very informative about the choice to be made, since they result in very similar ranges of values for particles of both classes. Moreover, in this case, the choice made from the sampled particle was the wrong one.

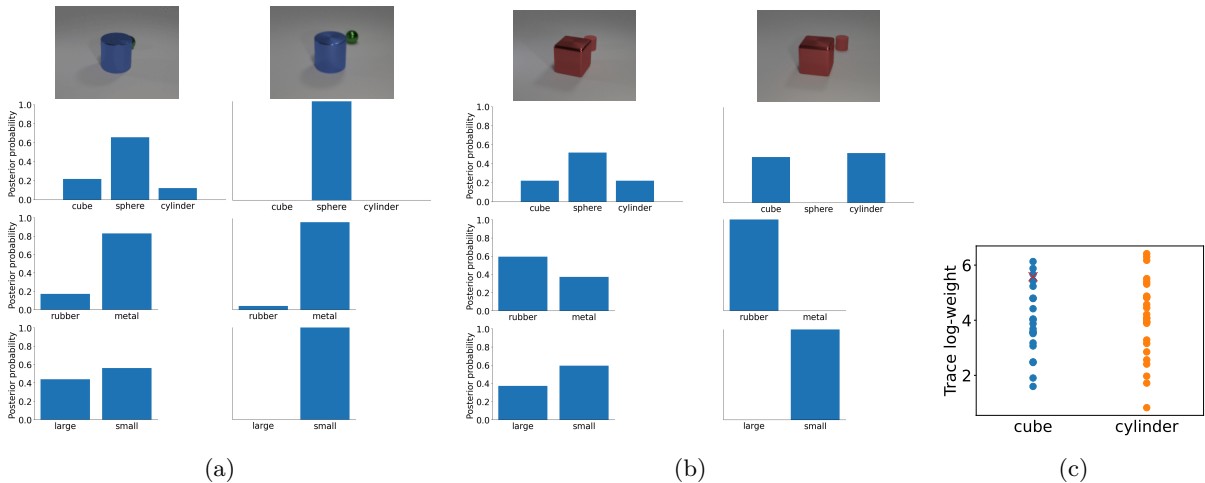

Figure 5: **(a)** Occlusion between objects of **different** colors and **(b)** between objects of the **same** color. From top to bottom rows, inferred properties represent object shape, material and size, respectively. **(c)** In the case where uncertainty remains high (image **(b)**), it might be the case that the resampled trace (marked as the red ×) selected to compute mAP results is not the one that better predicts the properties of observed objects.

## 5 Conclusions

We approached object-centric representation learning with an inference neural module composed by specialized proposal networks that learn to invert a probabilistic program for scene generation. By combining a state-of-the-art neural architecture with object-centric inductive biases with a symbolic generative program, ICSA exhibits superior data efficiency with competitive property prediction performance while enhancing interpretability. In particular, ICSA shows enhanced robustness under OOD situations, with consistent and more plausible inference results when observing scenes explained by values out of the training support.

Moreover, the particle-based inference nature of ICSA allows to seamlessly observe the uncertainty associated with the posterior distribution proposed for each latent variable. We also show that ICSA can scale to more complex data environments while still holding the same data efficiency benefits over state-of-the-art fully neural baselines.

Finally, ICSA can represent a foundational framework for future approaches within object-centric representation learning in video data of interacting objects, where an intuitive physics engine can be modeled to emulate the noisy expectations that humans exhibit on object physical behavior during childhood (Ullman et al., 2018; Ullman & Tenenbaum, 2020; Lake et al., 2017).

## 6 Limitations

A limitation inherent to learning inference proposals in a fixed generative model lies in the fact that discrete random variables require pre-specified possible values. Even though continuous relaxations can be employed in some cases, learning new categories as data is observed (Sablé-Meyer et al., 2022; Mills et al., 2023) is a much more plausible hypothesis that should be implemented in the future. ICSA proposes a view on object representation learning by initializing its generative model with a certain amount of innate knowledge: for instance, the visual properties of objects. However, less complex primitive knowledge could be the basis for learning perceptual modules of object representations, while at the same time, learning primitive compositions of different shapes or colors to progressively better explain observations. This way, the generative model could also be learned through program induction (Rule et al., 2020; Lew et al., 2023).

We highlight that further applicability to real-world settings is limited by two main assumptions, related to each other. It is quite direct to recognize that prior latent factorization, even though often assumed for disentanglement, or more formally in non-linear independent component analysis (ICA) literature (Khemakhem et al., 2020; Hyvärinen et al., 2023; Zheng et al., 2022), largely limits scalability when it comes to models of image representations. Entities in the world are inherently related to each other (Schölkopf et al., 2021), which naturally drives these models to emulate possible dependencies if true scene understanding is the main goal. ICSA is also limited by this, and having richer priors that include dependencies among generative variables is a natural extension for this work. In this case, these dependencies must be learned such that the model is able to extract causally disentangled representations of each entity in the image (Komanduri et al., 2024).

The second assumption — probably the strongest one — is related to the prior knowledge of the true generative model. We captured ICSA's behavior under specific misspecification scenarios (Sec. 4.2), showing that we can still provide robust predictions when the prior generative model is not expressive enough to explain a certain observation. However, even though such scenarios pose an interesting OOD assessment, these still fall under the same domain as the one used for training, which is not the case when aiming to infer object properties in more realistic scenes. A direct path for scaling ICSA requires modifying the prior generative model and its rendering function (as already done in previous related works (Smith et al., 2019; Kulkarni et al., 2015; Gothoskar et al., 2023)), which might come with costs at inference times due to the increased computational burden of rendering each posterior hypothesis for likelihood evaluation. Also, running inference over real-world data would require access to the true generative model, which is not feasible. For fully neural approaches, this is not an issue, since these can flexibly be adapted to different datasets with minor architectural changes. Nevertheless, we believe that new methods for likelihood evaluation — for instance, methods that move from pixel-based hypotheses evaluations to a validation based on higher-level information — between simulated and real images could significantly enhance the applicability of approaches like ICSA in realistic data environments. Without the explicit requirement for the true generative model, pushing the development of more human-like likelihood evaluation procedures may bring ICSA closer to work within more realistic settings. This possibility is aligned with a shift in the *analysis-by-synthesis* paradigm, but we believe that it represents a plausible hypothesis for true scene understanding, instead of requiring the precise generation of whatever is observed. In addition, as we motivated in our experiments, these approaches should be implemented in a way that aligns evaluation scores with the sampled choices that compose the posterior, such that, from the set of posterior hypotheses, the most likely ones contain predictions that are closer to what is observed in reality. Unfortunately, pixel-level metrics often miss this alignment.

We finally acknowledge the failing mode that occurs on set prediction tasks caused by the need to specify an identity order over proposed traces for objects to be evaluated within the SR inference algorithm. Even though we still consider evaluating the posterior of each object sequentially a plausible direction to follow, and show its advantages over evaluating full traces, the chosen ordering scheme does not completely prevent ambiguities and places a non-negligible impact on inference quality. Therefore, new methods should be explored to break these ambiguities.

**Acknowledgments**

This work is financed by National Funds through the Portuguese funding agency, FCT - Fundação para a Ciência e a Tecnologia, within project LA/P/0063/2020 (`https://doi.org/10.54499/LA/P/0063/2020`) and under the PhD grant 2021.05767.BD (`https://doi.org/10.54499/2021.05767.BD`), and by the European Union under the Horizon Europe Programme AI4LUNGS project (Grant Agreement No. 101080756).

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

## A    Appendix

### A.1    Implementation details

**Additional details on SA-MESH**   The SA algorithm proposed in (Locatello et al., 2020) maps an encoded global representation of the input image into a set of distributed slots using an attention mechanism. It starts by randomly initializing a fixed set of slots $\mathbf{s}^{N \times D_s}$, on which a linear projection is applied to produce associated queries $\mathbf{q}^{N \times D_s}$. The input representation is also projected into keys $\mathbf{k}^{D \times D_s}$ and values $\mathbf{v}^{D \times D_s}$ distributed representations, which are then used to compute the dot-product attention (Luong et al., 2015). Here, $N$ refers to the number of instantiated slots of dimension $D_s$, and $D$ the size of the flattened representation of the input. Then, a double normalization procedure encourages not only slots to compete to represent input information exclusively but also all partitions of the input to be encoded by the group of slots (Locatello et al., 2020). SA-MESH takes an initial cost matrix $C_{ik} = d(Q_i, K_j)$ from a distance metric $d$ (e.g. Euclidean distance) between the set of queries and keys. Then, Equations 4 and 5 are repeated for $T$ iterations, returning a final attention map $A^{(T)}$. Minimizing the entropy $H$ of the transport map at each iteration $t$ increases the sparsity of slot representations.

$$\text{MESH}(C) = \underset{C' \in \mathcal{V}(C)}{\arg \min} H(\text{sinkhorn}(C')) \tag{4}$$

$$A^{(t)} = \text{sinkhorn}(\text{MESH}(C)) \tag{5}$$

**Proposal networks**   As explained in Sec. 3, posterior proposals are obtained through separated networks, consisting on softmax-activated linear layers, learning attribute-wise projections from slot representations to the number of attribute classes. Given the convergence into object-wise slots, we empirically found that object attributes were quite simple to approximate without requiring deeper or wider classification modules. Also, since we assume that there is no statistical dependency among objects, these proposals are computed given the representation of a single corresponding slot.

**Finding object positions**   Training a neural network to obtain posterior proposals for each location latent variable was considered an unnecessary challenge. Hence, we relaxed this problem by leveraging each slot's attention mask. Similarly to previous attempts to relax object location inference (Watters et al., 2017; Kipf et al., 2021; Kim et al., 2023; Wu et al., 2022; Biza et al., 2023), we used a 2-channel absolute grid (for $x$- and $y$-coordinates) such that inferring the position of all objects only requires computing the dot-product ($\cdot$) between the attention masks $A^{(T)}$ and the absolute grid.The resulting coordinate point was then used as the mean of a fixed-noise Gaussian posterior over hypothesized locations.

### A.2    Set prediction

**SA-MESH training details**   For the experiments reported in Sec. 4.1, we trained SA-MESH for 10k training steps, ensuring convergence, with batches of 256 images, using the Adam optimizer with a learning rate of $1 \times 10^{-3}$ and remaining parameters as default. Slots were randomly initialized, instantiated with a maximum number of objects set to 17 (according to the generated synthetic data used for training), slot iterations $T = 3$ and MESH iterations set at 4.

**Scaling the number of objects**  We ran inference with the ICSA-IS procedure over scenes ranging from 10 to 20 objects, aiming to investigate how the model behaves when increasing the number of objects inside and outside the training support. For this, we trained a model exactly at the same settings as in Sec. 4.1, only setting $N = 15$, instead of 10. From the results in Fig. 6, it is possible to observe that AP values tend to slowly decrease as the number of objects approaches the maximum value $N = 15$ set for training; if we keep increasing it beyond that, the model's actually responds well to having to instantiate a higher number of slots to encode more than 15 objects, and performance stabilizes. Increasing beyond 20 starts to become infeasible considering the scene dimensions.

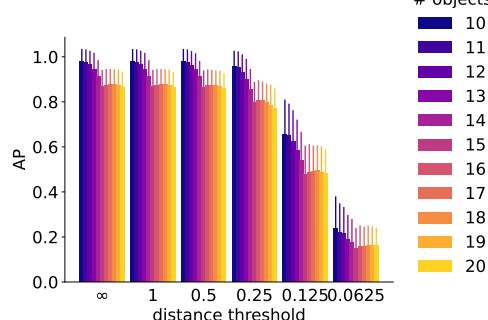

Figure 6: AP values obtained by running the ICSA-IS inference procedure over scenes with $N = \{10, \ldots, 20\}$ objects. During training, the maximum number of slots was set to 15.

**Additional inference results**  We also show comparative bar plots of each model used for object's property inference (ICSA-IS, ICSA-SR and SA-MESH) for increasing training power: in Fig. 9, all models were trained for 100 training steps, in Fig. 10 for 500 training steps, in Fig. 11 for 1000 and finally, AP performance of models trained for 10k training steps is shown in Fig. 12. An $\infty$ distance threshold denotes no distance criterion for an object to be considered a true positive once all predicted properties match the ground-truth. Note that, due to its sequential characteristic of the inference procedure, ICSA-SR property prediction performance holds at relatively the same overall values from higher to lower distance thresholds, while consistently decreasing when the number of objects present in the test scene grows. One of the reasons why this happens is related to object identity ambiguities in the sequential sampler: the more objects are to be inferred, the harder it becomes to account for the symmetries that result from ordering objects according to their Euclidean distance to the origin (see Fig. 13, in test images #2 and #3 for concrete examples). Hence, since proposal traces do not order the objects the same way, it is more likely that a certain object never gets explained in the final predicted scene. For instance, in test image #2 with $N = 10$, it is possible to observe that among all proposal traces, the small red square at the bottom was ranked at 4 different positions according to its predicted location. However, at these steps, different objects were selected to explain the observation according to the score-resample algorithm. In the last step, the SMC sampler only scored proposals for 2 objects, none of them corresponding to the "forgotten" one.

We took the set of test scenes with $N = 5$ and investigated how the log-likelihood of the set of particles evolves as the SR procedure goes through all objects to be inferred. Fig. 7 shows that as objects get explained, the averaged log-likelihood of the set of posterior proposals tends to decrease. Although this might seem counterproductive, i.e. one would expect that progressively explaining an image would result in increased log-likelihoods of the generated images at each inference step, but we found this is explained by small errors at location inference — propagated through the entire procedure — that happen to increase the error of selecting another object over selecting none. Regarding inference times, we computed for a single scene with 5 objects, and the relation between the set of inference strategies explored in this work is shown in Table 1. Surprisingly, the iterative nature of SR does not increase inference times by a significant amount when compared with IS, and SA-MESH holds the faster inference procedure as expected. This is explained by the fact that, in contrast with the experiments conducted with CLEVR, the image generation procedure in this data environment does not represent a computational bottleneck due to the simplicity of the data.

**Selecting the number of particles**  We ablate the impact of different amounts of inference particles, showing the resultant mAP values of the ICSA-IS inference procedure in $N = 5$ scenes, averaged across 5 random seeds (Fig. 8). We found consistent results over the different $\sigma$ values, with the main difference being showcased in the $\sigma = 0.0625$ case, where higher number of inference particles seem to have a larger positive impact in set prediction performance.

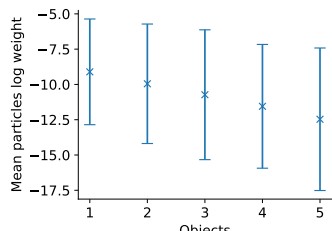

Figure 7: Log-likelihood weights averaged over the number of particles across $N = 5$ test scenes.

Table 1: Inference times of a single sample for each analysed inference strategy.

| Inference strategy | Time ($\times 10^3$ ms) |
|---|---|
| SR | 2.683 |
| IS | 2.502 |
| SA-MESH | 2.486 |

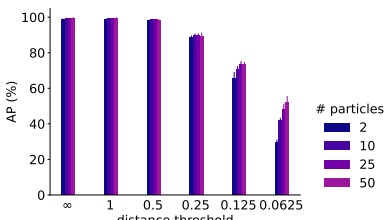

Figure 8: Ablation over the number of particles given by mAP values (%) averaged across 5 random seeds.

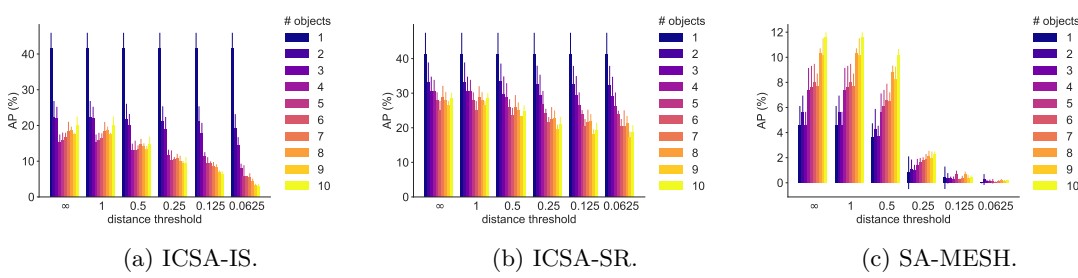

(a) ICSA-IS.  (b) ICSA-SR.  (c) SA-MESH.

Figure 9: Comparative plots of AP values at distance thresholds $\{\infty, 1.0, 0.5, 0.25, 0.125, 0.0625\}$ averaged over 5 random seeds. Vertical lines on top of each bar correspond to standard deviation values. Plots **(a)** and **(b)** correspond to the ICSA model with single-step importance sampling and following the score-resample procedure, respectively. Plot **(c)** shows the predictive performance of the trained SA-MESH model. All inference models were trained for **100** training steps.

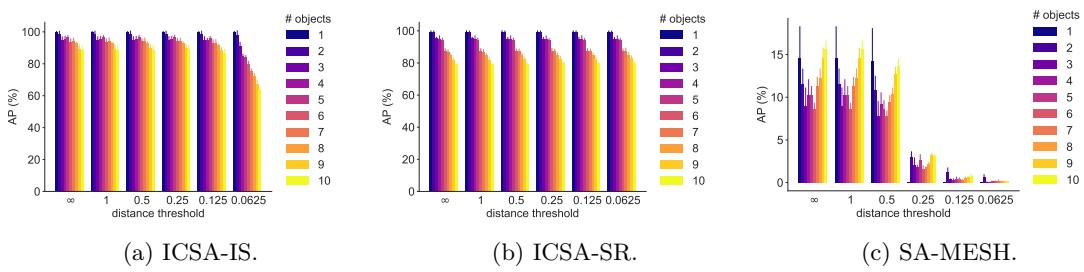

(a) ICSA-IS.  (b) ICSA-SR.  (c) SA-MESH.

Figure 10: Comparative plots of AP values at distance thresholds $\{\infty, 1.0, 0.5, 0.25, 0.125, 0.0625\}$ averaged over 5 random seeds. Vertical lines on top of each bar correspond to standard deviation values. Plots **(a)** and **(b)** correspond to the ICSA model with single-step importance sampling and following the score-resample procedure, respectively. Plot **(c)** shows the predictive performance of the trained SA-MESH model. All inference models were trained for **500** training steps.

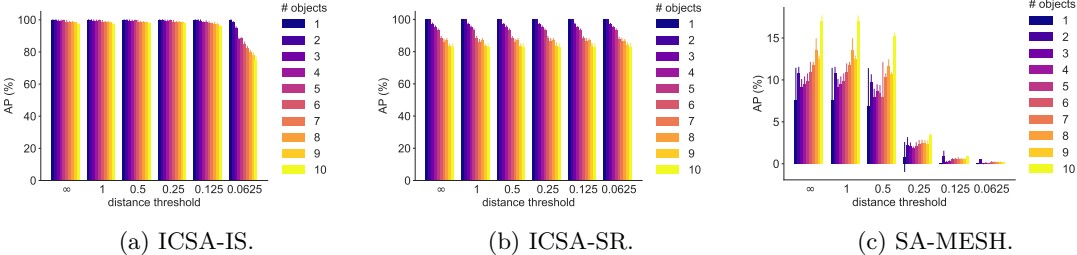

(a) ICSA-IS.  (b) ICSA-SR.  (c) SA-MESH.

Figure 11: Comparative plots of AP values at distance thresholds $\{\infty, 1.0, 0.5, 0.25, 0.125, 0.0625\}$ averaged over 5 random seeds. Vertical lines on top of each bar correspond to standard deviation values. Plots **(a)** and **(b)** correspond to the ICSA model with single-step importance sampling and following the score-resample procedure, respectively. Plot **(c)** shows the predictive performance of the trained SA-MESH model. All inference models were trained for **1000** training steps.

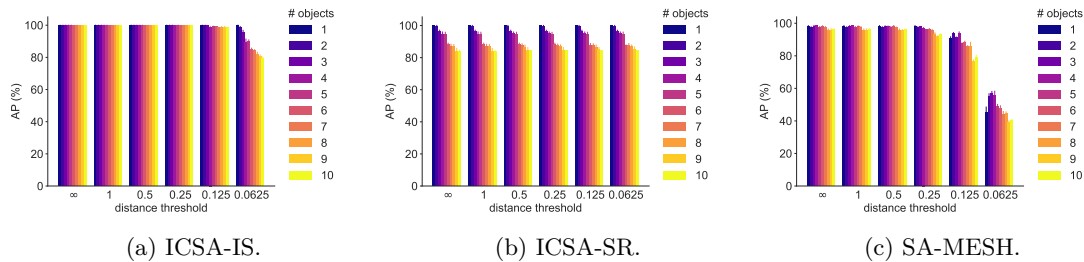

(a) ICSA-IS.  (b) ICSA-SR.  (c) SA-MESH.

Figure 12: Comparative plots of AP values at distance thresholds $\{\infty, 1.0, 0.5, 0.25, 0.125, 0.0625\}$ averaged over 5 random seeds. Vertical lines on top of each bar correspond to standard deviation values. Plots **(a)** and **(b)** correspond to the ICSA model with single-step importance sampling and following the score-resample procedure, respectively. Plot **(c)** shows the predictive performance of the trained SA-MESH model. All inference models were trained for **10000** training steps.

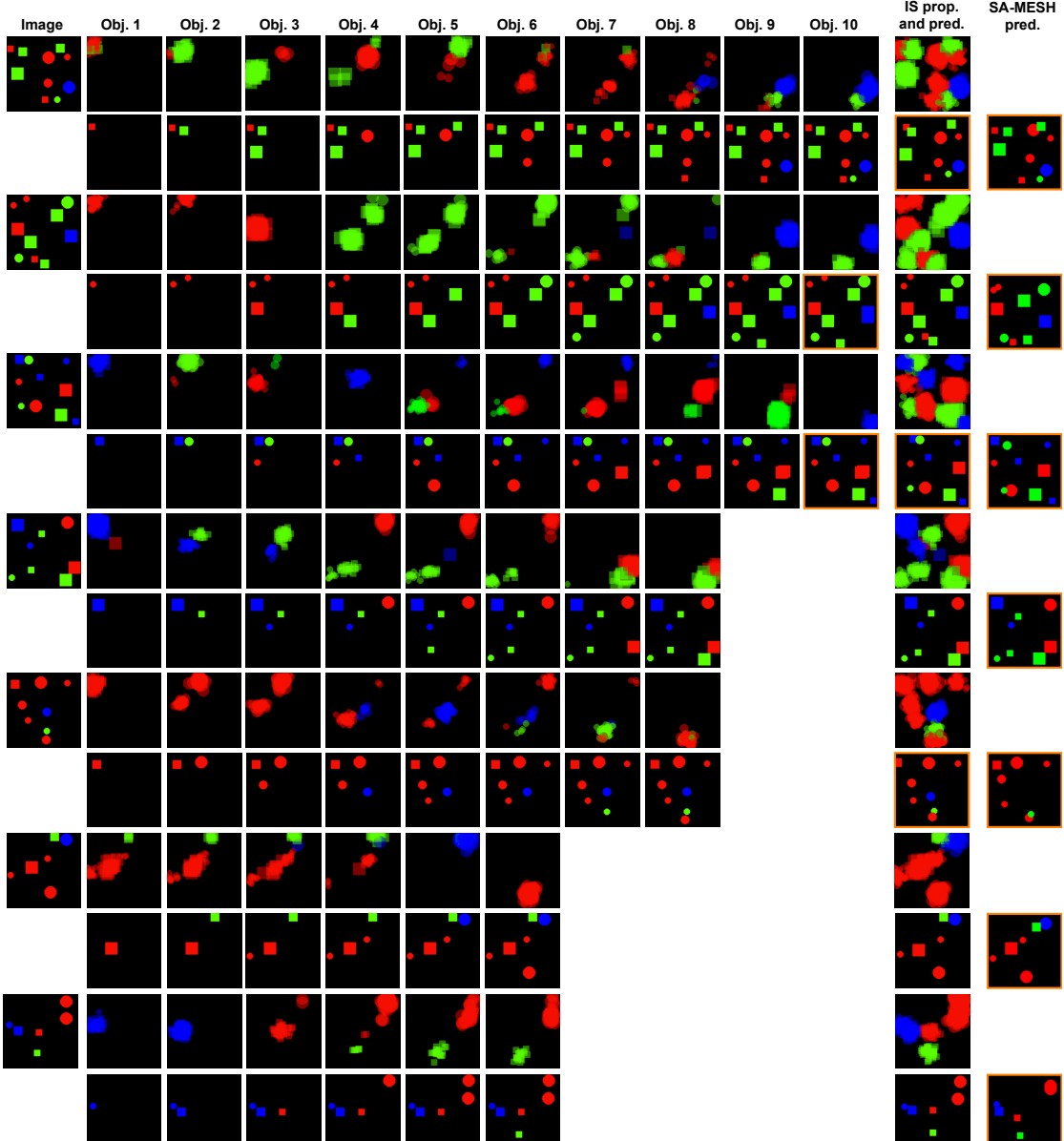

Figure 13: Examples showing the generated images resultant from each inference procedure, selected randomly among all test samples with $N \in \{10, 8, 6\}$. From left to right, the first column shows an observed test scene, followed by the sequential procedure of the score-resample inference of ICSA ($N$ columns, each one showing the overlay of proposal traces for a specific object on the top row, and the resampled proposal on the bottom row). Then, the following column shows the overlay of proposals to be weighted in IS inference (top row) and the resampled trace (bottom row). Finally, the last column shows the generated images when running the generative program with the object properties predicted by SA-MESH. All inference models were trained for **10000** training steps. Generated images with orange border denote examples where predicted locations cause objects to overlap (in ICSA-IS and SA-MESH) or ordering ambiguities cause missing an object (in ICSA-SR).

**CLEVR set prediction** For the experiments with CLEVR (Johnson et al., 2017), we maintained the same neural architecture, setting the batch size at 64 and the learning rate at 0.0004. Note that, since more latent variables are instantiated by the generative program, the neural module also appends the correspondent prediction heads (MLP). Also, in contrast with the procedure detailed in App. A.1, since CLEVR object positions cannot be directly inferred from attention masks, the posterior over their coordinates is also computed from specialized prediction networks.

