# OpenReview forum: "Learning object representations through amortized inference over probabilistic programs"
_TMLR — Accepted by TMLR_

### Review · Reviewer_gXPq · 2025-10-27

**Summary Of Contributions:**

This work proposes ICSA, a method that combines inference compilation with slot attention. The idea is that a neural network learns to predict posterior distribution for a fixed generative process (thus, in fact, learning to invert the generative process). This network can then be used at inference time to propose particles that are then fed into the prior generative program.

STRENGTHS
- The problem of combining symbolic information with slot attention (object centric learning in general) is very relevant and interesting
- The work despite being a bit incremental is novel.
- Experiments are quite narrow in domains, but well executed.
- Limitations are clearly stated.

WEAKNESSES
- The paper is written in a confusing and unclear way.
- The method seems quite ad-hoc for the experiments.
- Experiments are not very impressive (still insightful though).

FEEDBACK / QUESTIONS:
- Despite being quite familiar with the topics of the paper, it was very difficult to read and understand. Some points of improvements: avoid long sentences (there are a lot of sentences that are 3/4 lines long), avoid using too many synonyms for the same thing (moves the effort to the reader to connect the dots), use examples / intuitive explanations to explain difficult concepts (e.g. when explaining inference compilation, would be more useful to discuss a brief example instead of trying to explain the entire idea in abstract terms; the description of the latent discrete variables (shape, size, ..) is quite important and centra to the paper, it should be in the main body because it helps comprehension a lot), structure could be improved a bit (e.g in “Method” would be better to follow the same structure of Figure 1, so training and inference; in experiments might help to write at the beginning the experimental questions, so it’s clearer where’s the focus), if terms are used to clarify, they should be well-known or explained (e.g. the method sections starts with “Our proposed system follows the ideas from the analysis-by-synthesis paradigm and the Helmholtz machine (Hinton et al., 1995; Dayan et al., 1995)”, but unless someone is familiar with the terms, it just adds confusion and more questions). I will stop here with the feedback on the writing style, but I believe the text would benefit of a serious re-writing.
- For the method, a question: I did not understand how the proposals at inference time are then combined to get the final predicted image. Can you explain or point me to where it is discussed?
- In 3.1, does the generator have to be non-differentiable? It looks like the entire section is just to describe a rendering method, and Algorithm 1 seems to add more confusion than clarifying, unless I’m missing something…?
- In 3.2.1, most of it seems to be just MESH, shouldn’t it be in the preliminaries instead?
- 3.2.2 looks like an implementation detail, either explain better why it is fundamental, or it could easily be moved to the appendix.
- In 3.2.3, is where an example could really clarify things, but from my understanding this check of permutations is quite important. Although, it doesn’t seem very scalable, does it?
- In equation 5, the fact you assume to have access to that joint distribution at the numerator, seems quite a strong assumption. I guess it is what you also discuss in the limitations, but how do you have insights on how to relax it?
- In general, it seems that the advantage ICSA gets, is due to the heavy induction bias. This is often the case in neurosymbolic methods, but in this case it looks like the bias is “heavier” than usual. It’s really forcing the network to predict some discrete latent variables that then makes the goal of generating very easy. I am not sure how easy is it to adapt to other domains. It feels very ad-hoc, and I am not 100% convinced that the same techniques would easily translate to more complex scenarios. What are your insights on this?
- Experiments might benefit from an ablation study to isolate what is contributing most.
- In the conclusions it is mentioned that interpretability is enhanced. However, neurosymbolic methods are known to suffer from reasoning shortcuts, and I am wondering if it is possible that the network in your model learns to predict latent distributions that actually go against what would be the right interpretation, but that somehow pushes the relevant metric up?

**Audience:**

Yes

**Audience Explanation:**

As mentioned, the topic is relevant, interesting, and the proposed method is novel enoguh to make it interesting to share the insights provided here.

**Claims And Evidence:**

Yes

**Claims Explanation:**

Experiments support the claims made by the authors (even though it is easy to get lost in the text).

**Requested Changes:**

My biggest concern is the clarity of the text. If that can be improved, than all the rest is secondary.

---

> ### Author Response · Authors · 2025-11-24
> **Response to Reviewer gXPq - part 1**
>
> We thank the reviewer for the thorough reading of our paper and for the valuable comments. Below, we provide a point-by-point response to the concerns raised, being completely available for further discussion if necessary.
>
> **Questions:**
>
> > Despite being quite familiar with the topics of the paper, it was very difficult to read and understand.
>
> We reviewed the entire article, focusing on improving clarity and making the manuscript lighter for the reader. We also rearranged Section 3 according to Fig. 1 and added the full description of latent variables to Sec. 3.1.
>
> > In experiments might help to write at the beginning the experimental questions, so it’s clearer where’s the focus.
>
> We agree with the reviewer and added that information. In Sec. 4.1., we explained what is the main objective of the set prediction taks, as the name may not be truly self-explanatory. We also better framed the main reasons behind the experiments in Sec. 4.2., being in line with concerns of inference scalability in data not ruled by the same generative assumptions.
>
> > If terms are used to clarify, they should be well-known or explained…. unless someone is familiar with the terms, it just adds confusion and more questions.
>
> We clarified the example raised by the reviewer by explaining in the text why our approach is under the “analysis-by-synthesis” and the “Helmholtz machine” modelling classes. We still don't describe them in depth, but leave the reader with appropriate references; however, we now believe that the resemblance is clearer for any type of reader.
>
> > For the method, a question: I did not understand how the proposals at inference time are then combined to get the final predicted image. Can you explain or point me to where it is discussed?
>
> Regarding the evaluation of proposals at inference time, we explain this in Sec. 3.3. Essentially, a set of $K$ particles is sampled from the posterior distributions computed for each latent variable; then, each particle is rendered and the hypothetical scene is obtained for the likelihood assessment. There is only the difference in the score-resample alternative, which repeats this process in an object-wise manner, i.e., it generates and scores a single object at each inference step, until there are no objects left.
>
> > In 3.1, does the generator have to be non-differentiable? It looks like the entire section is just to describe a rendering method, and Algorithm 1 seems to add more confusion than clarifying, unless I’m missing something…?
>
> Sec. 3.1. describes not only the rendering method but also the generative model itself, explaining assumptions on latent factorization, their prior distributions and supports, as well as specific generative conditions assumed (e.g. no occlusions). Regarding Algorithm 1, given its simplicity, we recognize that it might not bring new information, and we have removed it.
>
> > In 3.2.1, most of it seems to be just MESH, shouldn’t it be in the preliminaries instead?
>
> Sec. 3.2.1 is essentially a high-level explanation of SA-MESH, indeed. We intentionally left this description within the “Methods” section so that the reader could find every core component of our inference module in a single section. However, we understand that some of the content in 3.2.1. is not actually crucial for understanding the method itself. This way, we moved the additional details into the appendix and left the core information regarding SA-MESH and the reasons behind its choice (i.e., multiset-equivariance ) in 3.2.1, in a more concise way. We hope this change helps to clarify the role of Sec. 3.2.1, providing only the essential information the reader will require in the following sections.
>
> > 3.2.2 looks like an implementation detail, either explain better why it is fundamental, or it could easily be moved to the appendix.
>
> We agree that subsection 3.2.2 is not fundamental for the core understanding of our approach. We moved it to the appendix.
>
> > In 3.2.3, is where an example could really clarify things, but from my understanding this check of permutations is quite important. Although, it doesn’t seem very scalable, does it?
>
> The object order fixing is quite important for the score-resample inference procedure, and it becomes less reliable as the number of objects in the scene grows. Since this procedure orders objects in a proposal trace according to their Euclidean distance (computed from their inferred location) relative to the origin of the image, it is more likely that objects end up being placed at identical distances in scenes with a higher number of objects. We kindly point the reviewer for the results in Sec. 4.1., where we discuss these ordering issues using an example from Fig. 13 (more concretely, in the second scene).

---

> ### Author Response · Authors · 2025-11-24
> **Response to Reviewer gXPq - part 2**
>
> > In equation 5, the fact you assume to have access to that joint distribution at the numerator, seems quite a strong assumption. I guess it is what you also discuss in the limitations, but how do you have insights on how to relax it?
>
> The reviewer is correct, this numerator in Eq. 5 represents the likelihood evaluation of each particle under the joint prior distribution. As discussed in “Limitations”, assuming access to a prior generative model is required for training our posterior proposal modules the way we do. In our view, relaxing this assumption would mean having a simpler generative model (i.e. not expressive enough to perfectly explain the data) alongside more complex likelihood evaluation methods such that correctly inferring object attributes could be achieved without having to precisely generate the observation; nevertheless, this view still requires a generative model to sample from.
>
> > It feels very ad-hoc, and I am not 100% convinced that the same techniques would easily translate to more complex scenarios. What are your insights on this?
>
> We recognize the importance of this concern, considering that, as is, there are certain assumptions that prevent the usage of an approach like ICSA in real-world settings. As we try to elaborate in Sec. 6, we argue that a shift in the analysis-by-synthesis learning paradigm would push the development of more human-like likelihood evaluation procedures, which may bring ICSA closer to working within more realistic settings (since the requirement for the true generative model of the data would be relaxed).
>
> > Experiments might benefit from an ablation study to isolate what is contributing most.
>
> We tried to cover a broad range of experiments and ablations to assess ICSA abilities at different extensions, but may the reviewer elaborate more on what kind of ablations could improve our contribution?
>
> > In the conclusions it is mentioned that interpretability is enhanced. However, neurosymbolic methods are known to suffer from reasoning shortcuts, and I am wondering if it is possible that the network in your model learns to predict latent distributions that actually go against what would be the right interpretation, but that somehow pushes the relevant metric up?
>
> We understand this concern to be somewhat closer to what we discuss in the “Results” paragraph of Sec. 4.3 (with the support of Fig. 4), where we reveal the misalignment that sometimes occurs between particle likelihood scores and the posterior samples that compose these, with “better” particles holding lower importance weights due to the pixel-wise likelihood score computation. In this case, we have a score metric that doesn’t completely ensure that hypotheses with higher scores are actually the ones we would think, ideally. This is why we stress this issue in our discussion in Sec. 6.

---

> > ### Comment · Reviewer_gXPq · 2025-11-24
> >
> > I'm afraid I was too sloppy when I recommended ablation study, and now I don't recall any more what I had in mind. I tried to re-check the paper, but I don't have notes specifically about the ablation. So, at this point I agree with your statement that you already do a pretty good job at it :)
> >
> > For the rest, thanks for working on the clarity, I believe the paper improved significantly!

---

### Review · Reviewer_eZWu · 2025-11-05

**Summary Of Contributions:**

# Method

(This is my understanding please do correct me it I am wrong)

ICSA proposes a neurosymbolic method for object-centric representation learning. The approach works as follows:

**Training process**: A generative model samples object properties (shape, color, position, etc.) from probability distributions and renders synthetic images. The neural network learns to invert this process - given an image, predict the object properties that likely generated it.
Architecture: Images are processed by a CNN encoder to extract spatial features. SA-MESH attention then runs iteratively (typically 3-5 iterations) to refine object-centric slots through competitive binding. In each iteration, slots compete to attend to different spatial regions of the image through a cross-attention mechanism. Queries come from slots, while keys and values come from image features. This iterative refinement allows slots to progressively bind to coherent image regions. Specialized neural networks take each slot as input and predict probability distributions for object properties with separate networks for each property type (one for shape prediction, one for color prediction, one for position, etc.).

**Training:** The system is trained end-to-end using KL divergence between predicted and true property distributions, which simplifies to cross-entropy for categorical variables (shape, color) and Gaussian negative log-likelihood for continuous variables (position).


**Inference:** At test time, given a new image, the trained networks extract slots using the same CNN encoder + SA-MESH process. The proposal networks then predict object property distributions for each slot. These predictions can be used directly, or refined using Sequential Monte Carlo methods with two approaches: (1) ICSA-IS propose complete scenes all at once, weight all hypotheses, and select the best; or (2) ICSA-SR sequentially discover objects one-by-one, scoring and resampling particles after adding each object until the scene is fully explained.


-----------------------

# Strength
- Novel neurosymbolic framework: ICSA is quite novel in that it uses SA-MESH as a component in a larger framework that connects neural visual processing with symbolic object representations through a fixed generative model.
- Training efficiency: Reports ~20x fewer training steps than SA-MESH baseline, demonstrating that the fixed generative model provides strong inductive biases and supervised learning is more data-efficient than unsupervised approaches.
- Out-of-distribution robustness: The OOD experiments are quite interesting as they show how the method can handle unseen data, maintaining coherent predictions when objects have properties never seen during training (new shapes, colors, or both simultaneously).
 -----------------------

#  Weaknesses:
- Clarity issues: The paper is difficult to follow and unnecessarily complex in its explanations. For example, the KL divergence discussion could simply state it's cross-entropy for categorical variables. Claims like "learning the generative model and the inference network at the same time makes this optimization loop highly complex" are made without empirical validation or testing.
- Unrealistic methodological assumptions: The approach relies on two major assumptions that severely limit real-world applicability: (A) the generative model is known a priori, which is almost never the case in practice, and (B) the visual world consists of a discrete, predefined set of object properties, which contradicts the infinite complexity of real visual scenes.
- Limited evaluation scope: The evaluation is restricted to overly simplistic synthetic datasets (basic geometric shapes) and CLEVR. Even on CLEVR, the method doesn't consistently outperform SA-MESH baselines on all metrics. No real-world data evaluation is provided, making it unclear how the approach would handle natural images or practical applications.

**Audience:**

Yes

**Audience Explanation:**

Despite its significant limitations, this paper would likely interest several segments of TMLR's audience:

- **Neurosymbolic learning researchers:** The combination of SA-MESH with probabilistic programming represents a novel approach that bridges neural and symbolic methods, which is an active area of research.

- **Object-centric learning community:** Anyone working on slot attention, object discovery, or structured representations would find the application of SA-MESH in this context interesting, even if the overall approach has limitations.

- **Inference compilation researchers:** The application of inference compilation techniques to computer vision problems is relatively uncommon and could inspire further work in this direction.

**Claims And Evidence:**

No

**Claims Explanation:**

- **Unvalidated theoretical claims:** Statements like "learning the generative model and the inference network at the same time makes this optimization loop highly complex" are presented as fact but never empirically tested. The authors don't demonstrate that their approach actually solves this supposed complexity issue.

- **Unfair efficiency comparisons:** The claimed 20x speedup over SA-MESH is misleading because it compares supervised learning (ICSA with ground truth labels) against unsupervised learning (SA-MESH without labels), making it an invalid comparison.

- **Unrealistic assumptions not addressed:** The paper requires known generative models and discrete property spaces, which severely limit real-world applicability, but these fundamental limitations are not adequately discussed.

- **Limited experimental validation:** Despite suggesting ICSA could work in complex settings, experiments are restricted to very simple synthetic datasets, providing insufficient evidence for broader claims about the method's capabilities.RetryTo run code, enable code execution and file creation in Settings > Capabilities.

**Requested Changes:**

- I feel the paper is lacking in terms of experimental results. One way to improve the paper is to conduct experiments on the same tasks as SA-MESH to enable fair comparison. Specifically, evaluate ICSA on Multi-dSprites and ClevrTex unsupervised object discovery tasks, and CLEVRER video experiments with varying object numbers. In addition, test the method on scenes with larger numbers of objects (beyond the current limit of 10) to demonstrate scalability and assess when the approach begins to break down.

- I think it would also be helpful to add a discussion of how the approach could be extended when the generative model is not known a priori, or when the world doesn't consist of discrete, predefined object properties. This is essential given that these assumptions severely limit real-world applicability.

---

> ### Author Response · Authors · 2025-11-24
> **Response to Reviewer eZWu - part 1**
>
> We thank the reviewer for the thorough reading of our paper and for the valuable comments. Below, we provide a point-by-point response to the concerns raised, being completely available for further discussion if necessary.
>
> **Weaknesses**
>
> > The KL divergence discussion could simply state it's cross-entropy for categorical variables.
>
> As explained in Sec. 3.2.3, the KL-div used in our approach is the inclusive KL-div, not the exclusive. The exclusive KL-div, in fact, reduces to the cross-entropy loss widely used in categorical prediction. In contrast, the inclusive KL-div flips its arguments, the approximated posterior becomes the denominator, which results in a broader density coverage since $q(z|x)$ must have non-zero density at every region where $p(z|x)$ has non-zero density too (something that is not encouraged in the exclusive KL-div / cross-entropy loss.
>
> > No real-world data evaluation is provided, making it unclear how the approach would handle natural images or practical applications.
>
> Our approach is focused on exploring the benefits of learning to invert a fixed generative model, demonstrating the advantages in terms of data efficiency, during training, and interpretability during inference. We recognize that such an approach is not a reliable choice to handle the complexity of real-world data, which is why we do not conduct any experiments in more realistic data environments. However, we have drawn some insights on how our assumptions could evolve so that approaches like ICSA could be closer to infer explanatory variables in more realistic data. We have now better addressed this issue in the "Limitations" section.
>
> **Unsupported claims**
>
> > Claims like "learning the generative model and the inference network at the same time makes this optimization loop highly complex" are made without empirical validation or testing.
>
> We agree that we present no evidence that validates this claim, and our experimental work is not designed to investigate this idea. Hence, we have removed it so as not to raise the expectation in the reader that experiments to support that claim will be found in the paper.
>
> > The claimed 20x speedup over SA-MESH is misleading because it compares supervised learning (ICSA with ground truth labels) against unsupervised learning (SA-MESH without labels), making it an invalid comparison.
>
> We would like to clarify that our experiments were all within supervised settings, both ICSA and SA-MESH. So, we are actually comparing both approaches under an identical experimental setup (the reviewer can now find this clarification in the introductory text of Sec. 4.1, explaining what the “set prediction” task is about).
>
> > The paper requires known generative models and discrete property spaces, which severely limit real-world applicability, but these fundamental limitations are not adequately discussed.
>
> We point the reviewer for the second paragraph of the "Limitations" section, where we discuss the assumption of knowing the generative model *a priori*, alongside our view as to how that could scale. However, we have improved our discussion on real-world applicability in Sec. 6. Regarding the concern about having discrete property spaces, we consider such a choice as a simplification to restrict the space of combinations and speed up slots convergence - something that is usual to find in object-centric research, and it resembles the human notion of objects and their properties, which tends to be very "approximated" [1].
>
> > Despite suggesting ICSA could work in complex settings, experiments are restricted to very simple synthetic datasets, providing insufficient evidence for broader claims about the method's capabilities.
>
> In our experiments, we try to capture and define the data environments our approach, as is, can successfully perform inference over object properties. We design the experiments in a toy dataset for proof-of-concept, and then assess the model in a more complex, still synthetic, dataset. At this point, we can only discuss our main assumptions and possible lines to follow for future improvements to handle progressively more complex and realistic data.
>
> **Requested changes**
>
> > Evaluate ICSA on Multi-dSprites and ClevrTex unsupervised object discovery tasks.
>
> We understand why the reviewer made these suggestions, but these would not go in line with the main purpose of ICSA as a framework for learning posterior proposals through inference compilation. As in any work that addresses set prediction, we require access to target object properties for training our inference network; hence, we cannot train ICSA under unsupervised settings.
>
> [1] Ullman, T. D., Spelke, E., Battaglia, P., & Tenenbaum, J. B. (2017). Mind games: Game engines as an architecture for intuitive physics. Trends in cognitive sciences, 21(9), 649-665.

---

> ### Author Response · Authors · 2025-11-24
> **Response to Reviewer eZWu - part 2**
>
> > Test the method on scenes with larger numbers of objects (beyond the current limit of 10) to demonstrate scalability and assess when the approach begins to break down.
>
> We added this experiment, evaluating the ICSA-IS inference procedure in scenes with 10-20 objects, having trained the inference network to instantiate up to 15 slots. The results, presented in Fig. 6, allow us to observe how the model behaves when increasing the number of objects inside and outside the training support. It suggests that AP values tend to slowly decrease as the number of objects approaches the maximum value $N=15$ set for training; if we keep increasing it beyond that, the model actually responds well to having to instantiate a higher number of slots to encode more than 15 objects, and performance stabilizes. Increasing beyond 20 starts to become infeasible, considering the scene dimensions.
>
> > I think it would also be helpful to add a discussion of how the approach could be extended when the generative model is not known a priori, or when the world doesn't consist of discrete, predefined object properties. This is essential given that these assumptions severely limit real-world applicability.
>
> We tried to capture this concern in our misspecification inference scenarios, where basically we show what happens when we have a generative model that is not expressive enough to explain the observed data. Even though there could be several other misspecification settings that we could investigate, we believe that such experiments are already in line with the reviewer's concern, but we have tried to improve the discussion on why these experiments are important for assessing the practicality of our approach, which happens to be related to the discussion on scalability.

---

### Review · Reviewer_oHj4 · 2025-11-12

**Summary Of Contributions:**

This paper concerns representing the data such as images, consisting of multiple objects with a semantic description (latent variables). This scenario can be expressed as a joint probability distribution (generative model) between the data and the latent variables. The paper suggests a method to calculate the posterior distribution of the latent variables, that can then be used in various downstream tasks. Following earlier studies, the paper presents a synthetic scenario consisting of multiple objects with certain shapes and colors and presents results in the set prediction task.

To avoid modeling at a pixel level, the paper employs the slot attention mechanism (previous work), which returns a set of tokens corresponding to different objects and instead trains a neural based posterior model conditioned on these output tokens. In the downstream task, an importance sampling mechanism is employed.

**Audience:**

Yes

**Audience Explanation:**

As mentioned by the authors, obtaining models that include symbolic elements is a popular goal, which allows more explainable models and better inductive bias due to the domain knowledge transferred by the symbolic representation. The paper contributes to this line of research.

**Broader Impact Concerns:**

I do not have any concerns regarding ethical implications or other broader impact factors.

**Claims And Evidence:**

No

**Claims Explanation:**

I am not sure what the claims of the paper are. If the goal is to have better posterior distributions, other baseline methods for posterior estimation should be studied. The comparison to slot-attention is somewhat unfair because the latter does not intend to exploit the the generative model, while the proposed method is inherently based on such knowledge.

The details of the methodology in the paper also remain unclear to me (see below), and it is difficult to say if the paper has a significant contribution in terms of its methodology.

**Requested Changes:**

My main concern is that the paper is difficult to read for multiple reasons: First, the paper relies on the reference papers to the extent that it  becomes difficult to understand its contribution without referring to other papers. It is not clearly mentioned which parts of the suggested method are the contributions of the paper. There are terms such as "posterior proposal" or "particles", the definition of which is not clear to me. Even algorithm 1 has subroutines that are not explained as far as I see. I think that in many places, using a mathematical equation is much clearer than explaining the procedure in plain text.

My next concern is about the contribution of the paper. I think that there are two parts, which can be considered the contribution. First, f^prop and second the importance sampling scheme. First of all neither of these steps are explained in detail. It is not clear to me how f^prop looks like. This is important because one may expect that the posterior distributions of the objects not to be factorizable (i.e. they are dependent) even if the prior is factorized. I think that the sampling scheme where the objects are added one by one emphasizes this fact. Hence, if each slot is predominantly related to an individual object, I do not expect that f^prop is factorized over slots, otherwise, it should not be so significant. The way that the importance weights are further used is also unclear to me and I am not sure what the actual output of this stage is. Is it a set of samples approximating the posterior, for example, which are updated by the importance coefficients? There seems to be also a contradiction between f^prop and the IS. If I am correct f^prop evaluates the probability density/mass function, while SA will sample it. So, which one is the goal?

Finally, as also mentioned in the limitations part, this work is limited by requiring the knowledge of prior. Although the authors examine some OOD experiments with unseen shapes/colors. In practice, the dependency of the object is perhaps more critical. For example, a crosswalk in a driving scene is perhaps strongly related to existence a pedestrian. In a neuro-symbolic scenario this relation is often only partially known. I am not sure if the proposed method can address such scenarios. Moreover, as the final stage depends on  statistical sampling, it may become highly complex in the presence of many inter-related objects (as an exponentially growing batch of samples will be needed).

---

> ### Author Response · Authors · 2025-11-24
> **Response to Reviewer oHj4 - part 1**
>
> We thank the reviewer for the thorough reading of our paper and for the valuable comments. Below, we provide a point-by-point response to the concerns raised, being completely available for further discussion if necessary.
>
> **Unsupported claims**
>
> > I am not sure what the claims of the paper are. If the goal is to have better posterior distributions, other baseline methods for posterior estimation should be studied. The comparison to slot-attention is somewhat unfair because the latter does not intend to exploit the generative model, while the proposed method is inherently based on such knowledge.
>
> The updated manuscript now contains a “Contributions” paragraph to summarise the main claims and research questions we tried to investigate in this work. Following a neurosymbolic line that argues in favor of having richer prior knowledge - in this case, in the form of a symbolic generative program - for learning image representations with neural networks, we approached this challenge under an object-centric scenario, demonstrating in practice the advantages in data efficiency and posterior uncertainty assessment. Regarding the direct comparison against SA-MESH, we believe that the fact that ICSA exploits the prior generative model is what allows us to draw conclusions about this main question.
>
> **Requested changes**
>
> > It is not clearly mentioned which parts of the suggested method are the contributions of the paper.
>
> We added a “Contributions” paragraph at the end of “Introduction” specifying the contributions detailed above.
>
> > There are terms such as "posterior proposal" or "particles", the definition of which is not clear to me.
>
> We will try to clarify these terms, and we have improved these definitions in the article to avoid any confusion for the reader. The term “posterior proposals” represent the proposed parameters for the posterior distribution of latent variables, computed after observing a certain sample, by specialized neural networks given an object-centric representation/slot (defined in the introduction of Sec. 3.2); the term “particles” represent a set of posterior hypotheses that are sampled from the proposed posterior distribution, which are further weighted/scored so that the model returns a final decision for the state of each latent variable (defined in Sec. 3.3).
>
> > Even algorithm 1 has subroutines that are not explained as far as I see. I think that in many places, using a mathematical equation is much clearer than explaining the procedure in plain text.
>
> We are not sure which subroutines the reviewer is considering here, maybe the procedure for checking possible object occlusions during sampling? Nevertheless, we found that Algorithm 1 was not giving the reader useful information, so we decided to remove it in the updated version of the manuscript. Regarding the suggestion of replacing plain text with mathematical formulation, we kindly ask the reviewer to point for examples where we could benefit from this alternative.
>
> > It is not clear to me how $f^\textnormal{prop}$ looks like. This is important because one may expect that the posterior distributions of the objects not to be factorizable (i.e., they are dependent) even if the prior is factorized. I think that the sampling scheme where the objects are added one by one emphasizes this fact.
>
> The $f^\textnormal{prop}$ proposal modules are softmax-activated linear layers that compute the posterior distribution for each latent variable, given a slot representation. We assume that there is no statistical dependency among objects; hence, these proposals are only computed given the representation of their corresponding objects. Regarding the relation of this assumption with the score-resample inference scheme, since it implements a sequential procedure for scoring posterior proposals associated with a single object at a time, this score may be affected by how that particle contributes to explain the whole scene, given already explained objects in previous inference steps. More concretely, while scoring a particle $k \in [K]$ for object $i \in [N]$, if in a previous inference step there was already accepted an object at the same location, the score for particle $k$ will probably be lower because it is just proposing to place an object where there is already one. So, there is this relation in particle scoring for this inference procedure, but the posterior estimation computed for object $I$ is not dependent on the estimation of object $j$, $j<i$.

---

> ### Author Response · Authors · 2025-11-24
> **Response to Reviewer oHj4 - part 2**
>
> > The way that the importance weights are further used is also unclear to me and I am not sure what the actual output of this stage is. Is it a set of samples approximating the posterior, for example, which are updated by the importance coefficients?
>
> We use the importance weights to further resample a single particle (for each object sequentially or for all objects at once, depending on the inference procedure) to be used in AP computation. Each particle $k$ holds a different posterior $q(\hat{z}^k|\mathbf{x})$ computed to explain the scene $\mathbf{x}$ - which reduces to the sum of the log-likelihoods of the posterior distribution proposed for each latent variable - but AP assessment requires sampling a single prediction for each latent variable.
>
> > There seems to be also a contradiction between f^prop and the IS. If I am correct f^prop evaluates the probability density/mass function, while SA will sample it. So, which one is the goal?
>
> We believe that this concern may result from some confusion on the $f^\textnormal{prop}$ modules, which, as mentioned above, propose the parameters for each posterior distribution, while IS evaluates the set of joint proposals to resample the particle that better explains the observed scene.
>
> > This work is limited by requiring the knowledge of prior. In practice, the dependency of the object is perhaps more critical. (…) I am not sure if the proposed method can address such scenarios.
>
> In fact, the proposed method cannot directly scale for real-world data that comes from arbitrarily complex data generative processes. However, is this concern related to the model’s generalization when the data comes from a low-density (possibly even zero) region of the prior? We agree with the reviewer that having richer priors that include possible dependencies among generative variables - as the crosswalk/pedestrian example, for instance - is a natural extension for this work. In this case, these dependencies must be learned such that the model is able to extract causally disentangled representations of each entity in the image [1], because it will (probably) fail if the posterior approximation disregards these possible dependencies and assumes an independent factorization of latent variables.
>
> > Moreover, as the final stage depends on statistical sampling, it may become highly complex in the presence of many inter-related objects.
>
> Assuming the existence of inter-related objects, we don’t expect increased complexity at inference time due to this, since the presence/absence of a certain entity would be directly computed from the inference network, and sampling is only required for scoring hypotheses and evaluating uncertainty. On the other hand, as mentioned in the previous response, we expect increased complexity in learning the representations of these inter-related objects since these dependencies should not be assumed nonexistent. Please note that these are only our intuitive insights on this idea, as we haven’t formally tested it.
>
> [1] Komanduri, A., Wu, Y., Chen, F., & Wu, X. (2024). Learning causally disentangled representations via the principle of independent causal mechanisms. In Proceedings of the Thirty-Third International Joint Conference on Artificial Intelligence (pp. 4308-4316).

---

> > ### Comment · Reviewer_oHj4 · 2025-12-19
> >
> > Many thanks for the response and revising the paper too. According to the authors explanation, I think that I have a better understanding of the contribution.

---

### Author Response · Authors · 2025-11-24
**Global Response to Reviewers**

We deeply thank the reviewers for their valuable feedback and constructive suggestions. We would like to highlight the main strengths recognized in our work, such as: (i) its novelty, by bringing together SA-MESH for slots extraction and structured inference for interpretability and uncertainty accounting, and (ii) the conducted experimental setup, not only emphasising training efficiency but also OOD robustness. Regarding the concerns expressed by the reviewers, we have carefully addressed them in the proper sections. We have already updated the manuscript, with the main changes marked in blue text. Below, we summarize the main modifications:

- **Clarification of contributions:** we added a “Contributions” paragraph, emphasizing the main contributions of our work.
- **Readability:** we reviewed the entire article to make it more clear and easier to read, by clarifying some core definitions that led to confusion, removing excessive adjectivation, and shortening long sentences to have more concise text; the structure of Sec. 3 was also modified to be aligned with the overall scheme given in Fig. 1.
- **Additional experiments:** from the suggestion of reviewer eZWu, we added an additional experiment by running inference on scenes with $N=\{10, \dots, 20\}$ objects, measuring not only the ability to handle more populated scenes but also the capacity of instantiating more slots at inference time (Fig. 6).
- **Scalability discussion:** we went deeper in the discussion about our insights on real-world scalability with ICSA, being a common concern among the reviewers.

Please note that all section references in the individual responses are made according to the modified manuscript.

---

### Decision · Action_Editor_P9dP · 2025-12-26

**Recommendation:** Accept with minor revision

**Additional Comments:**

Authors should carefully and explicitly address the final comments raised by Reviewer oHj4 in a revised version of the manuscript. In particular, the reviewer notes that overall structure remains difficult to follow for non-expert readers, and rely excessively on Figure 1. A clearer reorganization aimed at accessibility is therefore encouraged. Additionally, the reviewer also recommends incorporating more concrete details on the structure and role of the proposal network (f_{\text{prop}}), for example by integrating the clarifications already provided in your responses into Appendix A.1.

The reviewer also expresses a borderline assessment regarding the paper’s significance, noting that while the high-level idea appears novel, the current formulation—especially the per-slot application of (f_{\text{prop}}) and the resulting factorized posteriors—limits applicability to real-world scenarios. These limitations should be clearly acknowledged and appropriately reflected and discussed in the paper’s positioning and claims.

**Audience:**

Yes

**Audience Explanation:**

The paper addresses topics of clear interest to parts of the TMLR community, including researchers in neurosymbolic learning, object-centric representation learning, and probabilistic inference. Even within the stated modeling assumptions, the combination of slot attention with amortized inference over probabilistic programs provides insights that could be relevant to researchers working on structured representations, interpretability, and uncertainty-aware learning.

**Claims And Evidence:**

Yes

**Claims Explanation:**

The core claims are supported by the evidence presented in the work. This paper, as also noted by the reviewers, clearly specifies the proposed inference/learning pipeline and evaluates it on controlled object-centric benchmarks, showing competitive performance relative to the stated baselines under the same experimental setting and, also,  training-efficiency gains through targeted experiments (including an added scalability study with increased object counts). The remaining reviewer concerns are primarily about presentation/positioning and the scope of applicability (assumptions on the known generative program and factorization), rather than missing evidence for the claims.

---

> ### Author Response · Authors · 2026-01-13
> **Camera-ready submission**
>
> We would like to thank again to the Reviewers and the Action Editor for the valuable discussion.
> We have now submitted the camera-ready version of our manuscript, incorporating the points resulting from the discussion with reviewers and the additional comments raised by the Action Editor. More concretely, we think that the overall structure is easier to follow, and the final clarifications regarding the $f^{\textnormal{prop}}$ proposal networks were added in Appendix A.1. Also, an additional paragraph was added to discuss the limitation of assuming factorized latents in terms of scalability and real-world applicability.